# Impacts of agricultural machine renting on cereal crop productivity and commercialization in West Gojjam Zone, Ethiopia

Selam Tilahun[1]*, Berhanu Kuma[1], Amsalu Bedemo[2]

**1** Department of Agricultural Economics, Wolaita Sodo, University, Ethiopia, **2** Department of Policy Studies, Ethiopian Civil Service University, Ethiopia

* selam4998@gmail.com

## Abstract

Agricultural mechanization plays an essential role to increase production, productivity and commercialization of cereal crops. Despite the Ethiopian's government has made substantial efforts to increase production, productivity and commercialization of cereal crops by the introduction of selective and adaptable agricultural machines to smallholder farmers has remained low. This study, therefore, investigated the impacts of agricultural machine renting on cereal crop productivity and commercialization. The study used cross-sectional data collected from 192 agricultural machine users and 208 non-users from West Gojjam zone, Ethiopia. The data were analyzed using descriptive statistics and econometric models. The econometric model includes Trans log production function, Endogenous switch regression and propensity score matching. The result of descriptive statistics showed that the total factor productivity index of the cereal crop was 2.89, and the mean commercialization index for the sample households was 0.506. The estimation of results of Trans log production function model showed an average level of Technical Efficiency of 80.6%. The result of first stage Endogenous switch regression model showed that age, education level, ownership of oxen, total cultivated land, extension visit, access to information, family size, the position a farmer, and distance to the nearest market significantly affected adoption of agricultural machine renting. The result of endogenous switching regression and propensity score matching models showed that agricultural machine renting had a positive and significant impact on the selected outcome variables, total factor productivity, technical efficiency and output commercialization index. Based on the findings, the study suggests that the government and stakeholders should focus on strengthening the provision of education, development of infrastructures, extension service, to promote agricultural machine renting service, and enhance cereal crops productivity and commercialization.

**Data availability statement:** All relevant data are within the paper and its Supporting Information files.

**Funding:** The author(s) received no specific funding for this work.

**Competing interests:** The authors declare no conflict of interest.

## Introduction

Sub-Saharan Africa (SSA) is one of the regions in the world that is mainly characterized by smallholder farm households, where their livelihood depends primarily on agriculture. Similarly, Ethiopia's economy mainly depends on agriculture, which is the main source of income for 84% of the population, and it accounts about 33.8% of the country's Gross Domestic Product (GDP) and 70% of export earnings [1]. Cereal cultivation and marketing is the primary source of income for millions of subsistence farmers in the country. Cereals are accounts for 60 percent of rural employment in Ethiopia, 80% of cultivated land in the country, over 40% of food expenditure for the typical household, and over 60% of total calorie intake [2]. Cereals are the main food crop in terms of area coverage and production.

Cereal crop production and marketing have shown remarkable progress in Ethiopia, yet the domestic price for cereals is growing from time to time due to the rapid increase in domestic demand, rapid population growth, income, and urbanization [3]. Given the importance of cereal crops in national food security endeavors, smallholder farmers have used traditional methods of production with limited mechanization. Thus low productivity and, limited adoption of agricultural technologies including mechanizations and subsistence-based smallholder farming are characteristics of the sector [4,5]. Specifically, a low level of mechanization in agriculture continues to be the main obstacle to advancing cereal production, which, in turn, results in a high cost of production [6]. As a result, agricultural mechanization is among the mechanisms that increase production and productivity and reduce food insecurity [7].In the same way, according to [8] and [9] the utilization of appropriate agricultural mechanization technologies helps to increase the productivity and commercialization of cereal crops.

However, there is a public debates about the benefit of agricultural mechanization. According to the International Labor Organization (ILO, 1973) "mechanization leads to unemployment and is linked to economies of scale, meaning that it is more likely to be bought by larger farms [10,11]. In Africafew studies have conducted on impacts of mechanization on employment, and those that have mostly reported mixed effects [12–15] and positive effect [16,17]. According to [18] the impacts of mechanization on employment depend on location-specific factors such as wages, land availability, and type of mechanization. On the other hand, smallholder farmers can take advantage of mechanization by renting markets and cooperative exchange [19–21].

Developing and intensifying the cereal crops production has been the core of Ethiopia's agricultural development strategies and investments in the last 20 years to address the problem of low agricultural production and productivity [3]. Even though, still now, most of the interventions focused on the use of modern agronomic practice to improve agricultural productivity and production [22]. As a result, the problem has been continued with scanty achievement [8]. Many scholars state that the problem of low productivity is mainly related to low technical efficiency [3,23,24]. Thus, high productivity can be achieved by using appropriate agricultural machines with agronomic practice in the right manner [25]. Consequently, using appropriate and selective agricultural machines is the best strategy to increase production and productivity of crops, commercialization, food security [9,26].

Ethiopia's agriculture sector has depended on inexpensive and surplus labour for a long time. However, the condition is currently changing owning to the rapid migration of young people from rural to urban areas and the availability of universal access to education for all school-aged children [27,28]. This has led to a shortage of farm labor and pressure on the remaining household members. Additionally, the cost of oxen and lack of grazing land have become a major concern for smallholder farmers [19]. Consecquently, the use of machinery rental services has become a viable option for smallholder farmers. Recently, renting agricultural machines by smallholder farmers has been used in Ethiopia. Similarly, the Amhara region is one of the cereal crop growing areas and agriculture is the primary economic sector, but only 4.35% of agricultural activities oprate mechanically, which is far below the national average In Ethiopia farmers are familiar with the mechanization practice for the past 30–40 years, even though the index of agricultural mechanization was 12.1% in Oromia, 4.38% in SNNP, 4.35% in Amhara, and 3.48% in Tigray regions [29]. Likewise, West Gojjam zone is the major cereal crop production area across the nation [30]. However, utilization of agricultural machine with in this area remains minimal with farming practice relying on age-old and tradtional implements. Moreover, in recent times there has been increase in the population and a significant rise in average labor wage. This has lead to an increase the demand for agricultural machine renting among in small farmers in West Gojjam zone [31].

A number of studies have been conducted on the factors affecting the adoption of agricultural technology in Ethiopia; for example [32–37]. Only a few studies have been conducted on what may be the determinant of the adoption of the agricultural mechanization services with no specific studies on the agricultural machine renting services [19,38,39]. These previous studies applied logit, linear probability and tobit models to examine factors affecting agricultural mechanization. An important innovation in this paper was examining the impacts of agricultural machine renting on the productivity and commercialization of cereal crops. Hence there is no study on the impact of agricultural mechanization on productivity and commercialization. As result, this study provides information on smallholder farmers' agricultural machine renting and its impact on cereal crops productivity and commercialization which helps to draw a clear picture for policymakers involved in the development and dissemination of agricultural mechanization.

## Litrature review

### Theorthical review

Agricultural mechanization is encompassed within the broader concepts of agricultural transformation and the development of farming systems. According to [40,41], the demand for mechanized technology by farmers is a consequence of the process of agricultural intensification. [41] is well-known for establishing a theoretical and empirical connection between population growth, technological innovation, and agricultural intensification. In order to promote agricultural intensification, new technologies are developed and implemented, such as increased use of inputs like labor and fertilizer, as well as the integration of non-human sources of power such as animal traction or machinery.

Boserup's discussion of mechanization does not extend to consider the source of mechanized technology, nor the relative prices of land, labour, and outputs. The only effort in existing literature to conceptualize the circumstances under which farmers opt for mechanized technology, rather than agricultural intensification in general, is seen in the application of the induced innovation hypothesis by [42]. The demand for labor- or land-intensive technology by farmers is influenced by the relative costs of land, labor, capital, and output, which encourages public or private research institutions to innovate and adjust their development of new technologies to meet the needs of farmers. Agricultural intensification process needs mechanized technology, which is basically motivated by agro-ecological conditions, population density, and market demand. Where labor is relatively low, the wage of the agricultural labor will be high in comparison to the price of land. In such a situation, farmers will look for technologies that reduce labor-intensity in order to lower the overall cost per unit of production. Furthermore, the seasonal or spatial characteristics of agricultural production requires a series of specialized machines for land preparation, planting, weed control and harvesting – specifically designed for sequential operations, each of which is carried out for only a few days or weeks in each season.

## Empirical review

Different studies have been conducted on the determinants of the adoption of agricultural technology and its impact on productivity, cost of production and households' welfare. However, there is no specific study about the impacts of agricultural mechanization on crop productivity in Ethiopia.

According to a study regarding the effect of farm mechanization on productivity of rice farms in southern Ghana, the productivity of rice differs depending on the type of mechanization intensity. In this particular study, households who used mechanization from tillage, threshing, and transportation had the most significant and positive relationship with productivity. Moreover, land size cultivated, agrochemical expenditure, tillage intensity, education and transportation were significant factors that positively influenced productivity of rice [43]. [44] assessed the impact of farm mechanization on the productivity of wheat and maize crops in Pesh Awar valley. The result of Cobb-Douglas production function analysis revealed if the land of wheat and maize increases by one hectare, the productivity increases by 25.32 mounts for mechanized farmers to 23.22 mounds for non – mechanized farmers.

[13] examines the state, drivers and, consequently, the impacts of agricultural mechanization in eleven countries in Africa. Significant drivers of agricultural mechanization include the size of the household, gender of the household head, participation in off-farm economic activities, distance to the input and output markets, farm size, land tenure, type of farming system, access to extension services, and use of fertilizer and pesticides. His result showed that in general agricultural mechanization significantly raises the productivity of maize and rice in all cases. Similarly, [45] studied the impact of agricultural mechanization on agricultural production, income, and mechanism: evidence from Hubei province, China. This study analyzed the influence of agricultural mechanization level on agricultural production and income by utilizing a sample-modified endogenous merging model and a threshold effect model. The level of mechanization has a significant positive impact on the cost, output value, and income and return rate of all types of crops. For every 1% increase in the level of mechanization, the yields of all crops, grain crops and cash crops increase by 1.2, 1.56 and 0.43%, respectively.

## Conceptual framework

[40 and 41] explain agricultural intensification process needs mechanized technology, which is basically motivated by agro-ecological conditions, population density, and market demand. According to [20] demand for mechanization rose due to exogenous pushing factors of agricultural intensification, rising rural wages, urbanization, population pressure. Besides, rapid migration of the youth from rural to urban areas has a consequence of shortage of farm labor and cost draft animals changing the pattern of farming practice [19,27] Availability of machinery service provider in the nearby area of smallholder farmers is among the first factors that are contributing for making renting decision. Internal factors such as demographic and socio-economic and external factor like population growth, technological change, and development of infrastructure, market institutions and policy and are assumed to determine the renting decision of farmers for mechanization services. Finally, the adoption of agricultural machine renting had a positive impact on productivity and commercialization of cereal crops as shown Fig 1.

## Methodology

### Area description

The study was carried out in the West Gojjam Zone of the Amhara Region. West Gojjam Zone, is one of the eleven (11) administrative zones established under the Amhara region. The administrative zone is bordered to the north by the Northern and Southern Gonder zones; to the east by East Gojjam zone; to the south by Oromia region and to the west by Benshangul Gumuz region. Agriculture system in West Gojjam zone is a combination of both crop and animal farming activities.The primary crops caltitivated in the area include teff, wheat,maize, pepper, fingermillet, bareley beans peas chiechpeas,among others.In the region cearals occupy 84.41% of the total agricultural land (637552 hectares) and account for approximately 89% the total crop yield(19,580,577.34) quintals.

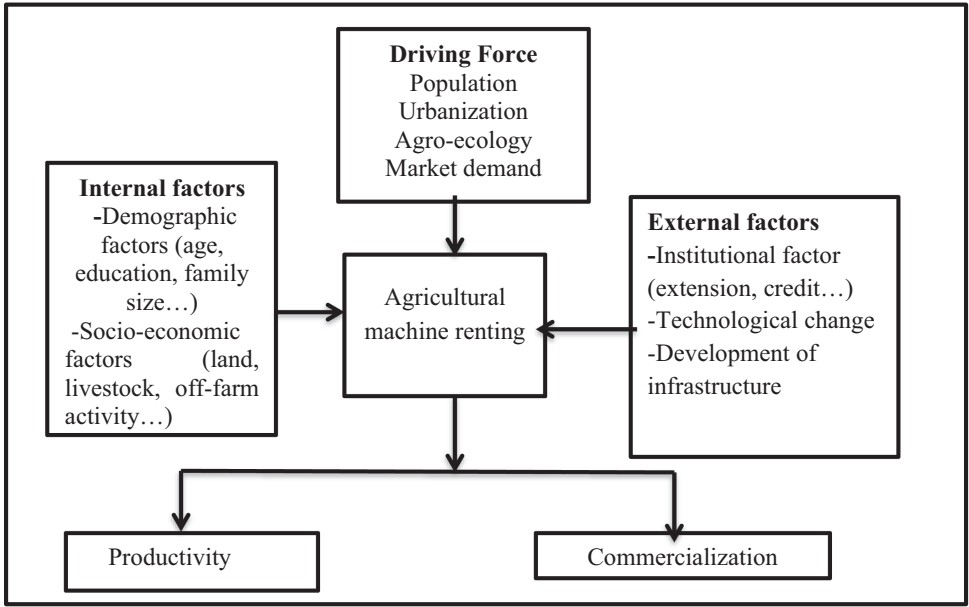

Source: based on literature [20, 27]

**Fig 1. Conceptual framework for impacts of agricultural machine renting.**

In the West Gojjam Zone, farmers use different agricultural machines for ploughing, harrowing harvesting and threshing through renting agricultural machines. The agricultural machine renting service (tractor, combine harvester, maize Sheller) was started in 1980 and the number of agricultural machines used by farmers increased from time to time. These machine renting services are provided by a union, an agricultural mechanization corporation and privately through communication and agreement with machine owners from the Oromiya region. The total number of agricultural machines in the west Gojjam zone in the union and agricultural mechanization corporation are 84, 10 and 180 tractors, combine harvesters, maize Sheller respectively. Agricultural machine renting payment was different according to the situation due to road, distance to the market (station) and season etc. An average fee for ploughing, harvesting (threshing and winnowing), especially for wheat, and shelling for maize, was 3500 birr/ha, 130–150 birr/qt, and 20–30 birr/qt respectively [46]

## Sampling procedure and sample size determination

A combination of purposive and multi-stage random sampling procedure was used to select districts, kebeles and sample households. Three districts from West Gojjam zone (Wonberma, Buriezuria and Dembecha) were selected purposively from the areas where agricultural machine rental practices took place. Then, in the first stage, a total of six kebeles were selected by simple random sampling method based on proportional to the number of Kebeles in each district. In the second stage households were stratified into two groups: those that adopted agricultural machines renting and those who did not. In the third stage, a representative sample size of 400 household heads were selected from both strata through systematic random sampling based on proportionate probability size of each districts and kebeles (145 households from Wonberma, 122 households from Burie Zuria and 133 households from Dembecha district)). Finally, 192 households that adopted at least one agricultural machine renting and 208 households that did not adopt an agricultural machine renting were obtained from six kebeles in the 2020/21 production season.as shown Table 1.

**Table 1. Selected districts, kebeles and their sample sizes.**

| Districts | Sampled Kebeles | Number of households | Adopters | Non-adopters | Adopter samples | Non-adopter samples |
|---|---|---|---|---|---|---|
| Wenberma | Markuma | 1029 | 492 | 537 | 33 | 36 |
|  | Koki | 1124 | 548 | 576 | 37 | 39 |
| Buriezuria | Alefa | 989 | 472 | 517 | 32 | 35 |
|  | Wadra | 808 | 397 | 411 | 27 | 28 |
| Dembecha | Wade | 659 | 308 | 351 | 21 | 24 |
|  | Asteboch | 1300 | 620 | 680 | 42 | 46 |
| Total |  | 5909 | 2836 | 3073 | 192 | 208 |

Source: Own computation results (data obtained from West Gojjam Zones Bureau of Agriculture and Rural Development, 2021)

[47] method was used to determine the sample size. Cochran's method is particularly suitable for stratified sampling designs, where the population is divided into distinct subgroups or strata, and is beneficial when the population size is large or unknown. The equation is written as:

$$n = \frac{z^2\, pq}{d^2}$$

(1)

where n is a minimum sample size, Z is 1.96 at 95% confidence level; p is smallholder farmers who may adopt in agricultural machine (50%); q is the weight variable and is computed as 1-p), and d is the desired precision or margin of error, expressed as a fraction of 0.05. Therefore, based on this formula, the sample size was supposed to be 385 farmers. For this study the data collected from 420 sample households by adding 5 household heads for each *kebeles* to avoid incomplete surveys or inaccurate data collected. Finally, the total sample size was 400 data were collected accurately (192 from machine renters and 208 from non- renters after discarding incomplete surveys or inaccurate data.

## Ethics approval

The ethical approval for the study was obtained from the "Wolaita Sodo University ethical board and College of agriculture" and the research was carried out based guidelines of Wolaita Sodo University ethics statement. Therefore, the study offered an approval number of 356/16 by the ethical committee. We also obtained a survey study titled "agricultural machine renting and its impact on cereal crop productivity and commercialization: the case of smallholder farmers in West Gojjam zone, Amhara region, Ethiopia". Informed consent was obtained from sampled smallholder farmers in the study area and all necessary data were collected through interview/verval/ method.

## Data types and sources

Both quantitative and qualitative data were used in this study. both secondary and primary data sources were used to generate data types. Primary data sources are farmers who rent agricultural machines and non-users of these machines. Secondary data sources include journals, books, CSA and internet browsing, national policies, and zonal, district and kebele reports.

## Data ollection methods

The major data collection methods included formal surveys, discussions with key informants and focus groups.The primary data were collected though face-to-face personal interviews using a semi-structured questionnaire. The data were collected based on the research objectives. The questionnaire included both open and closed-ended questions to collect

information relevant to the scope of the study. The questionnaire was first tested prior to the field survey for suitability (clarity, suitability and sequence of questions) and then re-tested based on the pre-test feedback.. In addition to interviews, discussion with each of the six selected kebeles extension agents who are working related to agricultural mechanization service (mechanization experts) were disscused.

## Methods of data analyses

Two types of data analysis methods were used to addressthe rereasch objectives: descriptive statistics and econometric models.

## Descriptive statistics

Descriptive analysis was used to analyze demographic, institutional and socio-economic characteristics of farm households. Further, descriptive statistics such as mean, proportion and standard deviation were used to describe economic and physical characteristics associated with households categorized based on their agricultural machine renting of mechanization behaviors. The independent sample t-test and chi-square tests were used to make comparisons between the categories with respect to key variables specified.

## Measurement of cereal crop productivity

In this study, TFP is the ratio of output to the total variable cost (TVC). This approach Total Fixed Cost (TFC), does not role since it is constant and does not have any effects the on the profit maximization and resource-use efficiency conditions [48]. According to [49] and [50] TFP for cross-sectional data or data within a specified time can be measured as the inverse of the unit variable cost because TFP is the ratio of the output and the Total Variable Cost (TVC) as shown in this equation;

$$TFP = \frac{Y}{TVC} = \frac{\sum_{i=1}^{n} Y}{\sum_{j=1}^{n} PijXji}$$

(2)

$$TFP = \frac{Y}{TVC} = \frac{1}{AVC}$$

Where, Yi: The output of cereal crops (in value or in quantities);

TVCi: Total Variable Cost of farm i;

Pji: Unit price of jth variable input used in farm i;

Xji: Quantity of jth variable input used in farm i.

## Measurement of market participation

**Measuring Crop Output Market Participation Index (COMPI):**. Agricultural commercialization is the proportion of agricultural production that is marketed and the households that sell more proportions of his/her product could be considered as commercialized farmer [51]. The meaning of commercialization goes beyond supplying output product to the market and it has to consider both the input and output sides of production [52]. In line with this, smallholder commercialization occur when farmers participate in agricultural markets either as a buyer (input) and/or seller (output) [53]. In this paper focused on output side of commercialization (market participation). Many empirical researches conducted on commercialization level crops [39,54–56]. At the household level, one definition of agricultural commercialization is defined as the ratio of the value of agricultural outputs sold to the total value of agricultural outputs produced by a household [57–59]. Hence, output side commercialization index or market participation for a household (COMPI) can be computed as:

$$COMPL_K = \sum_{i=1}^{N} \frac{S_{ki}}{Q_{ki}} \quad ; \quad Q_{ki} \geq S_{ki}$$

(3)

Where, $COMPI_k$ is the proportion of crop k sold ($S_{ki}$) to the total amount produced ($Q_{ki}$) aggregated over the total sample households in a given location.

## Efficiency and productivity analysis

The study used the production data of cereal crops (teff, wheat and maize) in the 2020/21 production season. The stochastic frontier methodology was later developed by [60,61]. It is specified as

$$y_i = f(x_{i,}\beta) + \varepsilon_i(v_i - u_i)$$

(4)

Where: Yi is the amount of the cereal crops yield produced by ith household in the 2020/21 production year that is expressed in quintals; is a production function$f(x_i, \beta)$are the parameters to be estimated; X's are exogenous input factors (labour, land, fertilizer seed and herbicide) affecting the amount of yield; $\varepsilon_i$is the error term, equal to ($Vi - Ui$); $Vi$ is a two-sided random error component beyond the control of the farmer, and $Ui$is one-sided inefficiency component.

Given the level of input, the TE of the $i^{th}$ cereal crop producer farmers are the ratio of the observed output to the maximum potential output (frontier output) [60].

$$TE_i = \frac{actual\ output}{potential\ output} = \frac{f(x_i; \beta)\exp(v_i - \mu_i)}{f(x_i; \beta)\exp(v_i)} = \exp(-\mu_i)$$

## The impacts of agricultural machines renting on productivity and commercialization

To measure the impact of agricultural mechanization on the productivity and commercialization of cereal crops, endogenous switch regression (ESR) and propensity score matching (PSM) models were employed.

When assessing the impact of adopting an agricultural machine on the productivity commercialization of a farm household, it is important to capture both the observable and unobserved characteristics of the adopter (treatment group) as well as the non-adopter (control group). However, the majority of impact assessment methods that use non-experimental data do not capture the observable and/or non-observable characteristics that influence adoption as well as outcome variables. For example, instrumental variables only capture unobserved heterogeneity but assume that the parallel shift in outcome variables can be seen as a treatment effect [62,63,64].

In contrast, using regression models to analyze the impact of a given technology using pooled samples of users and non-users might be inappropriate because it gives the similar effect on both groups [64]. A methodological approach that overcomes the aforementioned limitations is ESR, which is the most frequently used common method to analyze the impact of a given technology [64–66] In this paper, we employ parametric ESR with non-parametric PSM technique to reduce the selection bias and assure consistent results by capturing both the observed and unobserved heterogeneity that influence the outcome variable as well as the adoption decision.

The decision to adopt a technology can be modeled in a random utility framework by expressing the unobservable utility from adoption and non-adoption through observable variables [67].We specify the selection equation for technology adoption as:

$$\boldsymbol{P_i^*} = Zi\beta + U_i \quad P = \begin{cases} 1, & if\ p* > 1 \\ 0, & otherwise \end{cases}$$

(5)

Where $P_i^*$: is unobservable or latent variable for machine renting, Yi is its observable counterpart (the dependent variable use of agricultural machines equals one, if the farmer has adopted at least one agricultural machine during 2020/21 cropping season, and zero otherwise), Zi are non-stochastic vectors of observed farm and non-farm characteristics determining adoption and Ui is random disturbances associated with renting of agricultural machine of improved technology.

Farmers use at least one of the agricultural machines (tractor, combine harvester, and maize Sheller) in their farming activities. However, according [66] if the selection equation (first stage) is endogenous in the outcome equation (second stage), the result would be biased and inefficient. Therefore, it is vital to use instrumental variable methods to identify the second stage equation from the first stage equation. The instrumental variable should affect the adoption of agricultural machines renting but not the outcome variables, such as productivity and commercialization. The study use access to information about agricultural machine (yes = 1) as a selection instrument. Thus, this variable is more likely to be correlated with the adoption of agricultural machines, but not productivity and commercialization outcome variables or correlated with the unobserved. Moreover, we also check the validity of the instrument variable using a falsification test. The test shows that the variable significantly affects the adoption decision but not our outcome variables.

The two regimes that the smallholder farmers fall in to are represented by the following two regression equations.

$$Regime\ 1: \ Y_l = \alpha_1 X_i + e_{1i} \qquad if\ p = 1$$

$$Regime\ 2: \ Y_2 = \alpha_2 X_i + e_{2i} \qquad if\ p = 0 \tag{7}$$

Y1i and Y2i are the dependent outcome variables determined by the exogenous variables Xi, α1, and α2, are parameters that show the direction and strength of the relation between the outcome variable and the independent variables. $e_{1i}$ and $e_{2i}$ are error terms.

Finally, the error terms are assumed to have a trivariate normal distribution, with zero mean and non-singular covariance matrix expressed as:

$$cov\,(u_i,\ e_{1i},\ e_{2i}) = \begin{bmatrix} \sigma^2 u & . & . \\ \sigma^2 e_1 u & \sigma^2 e_1 & . \\ \sigma^2 e_2 u & . & \sigma^2 e_2 \end{bmatrix} \tag{8}$$

Where $\sigma^2 u$ variance of the error term in the selection equation, $\sigma^2 e_1$ and $\sigma^2 e_2$ are variances of the error terms in the continuous equations. $\sigma^2 e_1 u$ and $\sigma^2 e_2 u$ are covariance of $u_i$, and $e_{1i}$ and $e_{2i}$ respectively. Since $Y_{1i}$ and $Y_{21}$ are not observed simultaneously a covariance of the corresponding error terms is not defined [68]. This structure of the error terms indicates that the error terms of the outcome equation and the error term of the selection equation are correlated which results in non-zero expected value of $e_{1i}$ and $e_{2i}$ given $ui$ – error term of the selection equation [65]. Therefore, the expected values of the truncated error terms E($e_1$ | p = 1) and E($e_2$ | p = 0) are given below:

$$E\,(e_1\,|\,p\,=\,1)\,=E(e_1\,|\,U\,>\,-\,Z\beta)$$

$$+\,=\sigma e_1 u \frac{\phi(\frac{Z\beta}{\sigma})}{\Phi\frac{Z\beta}{\sigma}} = \sigma e_1 u \lambda_1 \tag{9}$$

$$E\left(e_2 \mid p = 1\right) = E(e_2 \mid U \leq -Z\beta) = \sigma e_1 u \frac{-\phi\left(\frac{Z\beta}{\sigma}\right)}{1 - \Phi\left(\frac{Z\beta}{\sigma}\right)} = \sigma e_2 u \lambda_2 \tag{10}$$

φ and Φ are the probability density and cumulative distribution function of the standard normal distribution, respectively. The ratio of φ and Φ evaluated at Zβ is referred to as the inverse Mills ratio λ1 and λ2 (selectivity terms). If the estimated covariance $\sigma^2 \varepsilon 1 u$ and $\sigma^2 \varepsilon 2 u$ are significantly different from 0 the decision to renting agricultural machine and the outcome variable (productivity and commercialization) are correlated. This implies endogenous switching and the presence of a sample selectivity bias [69]. Further, estimations of treatment effects will be made. Average Treatment effect on the Treated and Untreated (ATT and ATU) are computed using the results for expected values of the dependent variable for users and non-users in actual and counterfactual scenarios:

$$E\left(Y_{1i} \mid p_i = 1, X_{1i}\right) = \alpha_1 X_{1i} + \sigma e_1 u \rho_1 \frac{\varphi(z\beta)}{\Phi(z\beta)} \tag{11}$$

$$E\left(Y_{2i} \mid p_i = 0, X_{2i}\right) = \alpha_2 X_{2i} - \sigma e_2 u \rho_2 \frac{\varphi(z\beta)}{(1 - \Phi(z\beta))} \tag{12}$$

$$E\left(Y_{2i} \mid p_i = 1, X_{1i}\right) = \alpha_2 X_{1i} + \sigma e_2 u \rho_2 \frac{\varphi(z\beta)}{\Phi(z\beta)} \tag{13}$$

$$E\left(Y_{1i} \mid p_i = 0, X_{2i}\right) = \alpha_2 X_{2i} - \sigma e_1 u \rho_2 \frac{\varphi z\beta}{(1 - \Phi(z\beta))} \tag{14}$$

ATT is the difference between the expected value of the outcome variable from Equations 11 and 13; this is the difference between the expected value of the dependent variable for users and if they were not used. ATU is the difference between Equations 12 and 14 which estimates the difference between the expected value of the outcome variable for non-users of agricultural machine.

PSM techniques were applied to supplement the ESR model and to assess the consistency of the results with different assumptions such as nearest neighbour matching, caliper and radius matching, and kernel and local linear matching [70]. The treatment effects will be estimated based on matching estimators selected on the common support region [71]. The average treatment effects can be estimated using the inverse propensity weighing estimates as stated in IPSW [72] using matching techniques of Kernel Matching (KM), Nearest Neighbor Matching (NNM) and Radius Caliper Matching (RCM).

The selection of a particular approach depends on the data in question and, in particular, the degree of similarity between the treatment and control groups in terms of propensity score [73]. The Average Treatment Effect of an individual i can be written as:

$$\text{ATE} = E\left(Y_i^1 \mid D_i = 1\right) - E\left(Y_i^0 \mid D_i = 0\right) \tag{15}$$

Where ATE, Average Treatment Effect, which is the effect of treatment on the outcome variables:
$E(Y_i^1 \mid D_i = 1)$: Average outcomes for individual, with user, if he/she would participant
$E(Y_i^0 \mid D_i = 0)$: Average outcome of non-user, when he/she would non-participant

## Variables Definition and Hypotheses

**Adoption of agricultural machine renting:** A dichotomous dependent variable measured in participation of agricultural machine renting, taking value 1 if there is participation in agricultural machine renting and 0 otherwise.

### The explanatory variables

**Age of household head:** Is a continuous explanatory variable that measures the number of years of the household head. According to [74] when the household age has increased, the probability of technology adoption decreased. This is as a result of limited planning horizons when households are getting old. As a result, age of the household is hypothesized to affect agricultural technology adoption (tractor, combine harvester and maize Sheller) negatively.

**Family size:** It is the total number of family members, which is measured in terms of adult equivalent. Larger family sizes would be expected to decrease the probability that a farmer will use a combine harvester [38]. Consequently, family size would have a negative influence on the adoption of agricultural machines renting.

**Education level of household head**: Is a continuous independent variable measured in number of years attended formal education. Educated farmers would have the ability to perceive, interpret and respond to new information much faster than their counterparts without education [75] As a result, education is expected to correlate positively with the adoption of agricultural renting machines.

**Total cultivated land**: is a continuous variable and measured in hectares. It is frequently argued that farmers with larger farms are more likely to adopt improved technology (especially modern varieties) compared with those with small farms. According to [76] farm size of cultivated area of land has a significant influence on technology adoption decisions for wheat production. Therefore, farm size is expected to affect the adoption of agricultural machines renting positively.

**Credit use:** It is a dummy explanatory variable that takes the value of one if a household head has use of credit service and zero otherwise. Credits are expected to enhance farmers' financial capacity to purchase productivity enhancing. Use of credit services has a significant influence on technology adoption of wheat production [77]. So, a farmer who uses credit would have a positive influence on the adoption of agricultural machines.

**Extension contact:** This refers to the number of contacts per year that the household made with extension agents, and it is a continuous variable. The effort to disseminate new agricultural technologies is within the field of communication between the extension agent and the farmers at the grass root level [78]. Here, the frequency of contact between the extension agent and the farmers is hypothesized to be the potential force which accelerates the effective dissemination of adequate agricultural information to the farmers, thereby enhancing farmers' decision to adopt new crop technologies. As a result, frequent contact with extension agent would have a positive influence on the adoption of agricultural machines.

**Access to information**: It is measured as a dummy variable taking a value of one if the farmer had access to agricultural machine information from an extension agent, cooperative and farmers and zero otherwise. Farmers who had information from extension agents were more likely to adopt soil and water conservation practice [79]. Better agricultural machine information would help farmers to get detailed information about benefit, renting cost, source of agricultural machine.

**Off-farm activity participation**: This is a dummy explanatory variable that takes a value of one if a household head participated in off-farm activities and zero otherwise. Households with higher off-farm income may reduce the time allocated to farm management and resulting in lower productivity [80]. Therefore, participation in off-farm activity would have negative influence on the adoption of agricultural machines.

**Distance to the nearest market**: It is measured in kilometers. Distance to the nearest market is likely to influence the adoption of agricultural machines. The closer they are to the nearest market (station), the more likely that the farmers will receive valuable information and get machines. Farmers that are located far away from the nearest markets are less likely to adopt farm inputs than farmers who live near to market due to transaction cost [54]

**Distance from main road**: It is a continuous explanatory variable and measured in kilometers. Distance to the main road is used as a proxy for accessibility of transportation to markets. Distance from main road far from farmers residence renting of agricultural machines more expensive since due to transport cost and time [19]. Hence, households far away from roads are expected to be less probability of machine adoption.

**Distance to the development agent**: It is measured in kilometers. Proximity to development agents' office has the advantage of farmers gaining awareness and expose to new ideas and information about productivity inputs and outputs [81]. Households residing in places far from development agents' office were less likely to adopt agricultural machine than those who live near to development agents' office because of higher transaction costs, or may be information gap. This might be the cost of agricultural machines directly expressed in terms of distance from the station. Thus, the adoption status of agricultural machines would be reduced.

**Position of farmer in farmer's association:** it is a dummy variable coded as 1 if farmers have a position in rural kebeles, or model farmers and, 0 otherwise. Farmers who have some position in rural kebeles or model farmers are more likely to be aware of new practices as they are easily exposed to information [82]. Therefore, farmers who have a position in the association are hypothesized to have a positive relation with the adoption of agricultural machines renting.

**Number of oxen:** It is a continuous variable measured as the number of oxen owned by the household heads. All agricultural tasks are not performed only by agricultural machines renting; it also needs other additional oxen [83]. Thus, ownership of oxen has a positive influence on agricultural machine renting adoption.

**Livestock (excluding oxen)**: is a continuous variable measured in terms of Tropical Livestock Units (TLU) of livestock owned by the household excluding oxen. Livestock is the farmers' important source of income, food and draft power for crop cultivation in Ethiopian agriculture. It was hypothesized that as livestock ownership increases, adoption/intensity of adoption is expected to increase because it serves as proxy for wealth status [82].

**Membership to cooperative:** It is a dummy variable is measured by allocating a score of 1 if a farmer is a member of an organization and 0 otherwise. Belonging to a member of a cooperative influences a farmer's decision to adopt an improved technology. Being a member of the cooperative has a significant and positive impact on the adoption of teff row planting [84]. Therefore, farmers' membership of a cooperative would have a positive influence on the adoption of agricultural machines.

### Outcome variable

**Total factor productivity**: Total factor productivity index of cereal crops includes (teff, wheat and maize) produced by households in the 2020/21 production year.

**Technical efficiency:** Technical efficiency of cereal crops (teff, wheat and maize) in the 2020/21 production season.

**Output commercialization**: Cereal crop output commercialization indices include teff, wheat and maize output commercialization indicesin the 2020/21 production season.

## Results and discussions

### Demographic, socio-economic and institutional characteristics of households

The results statistical summary that shows the difference between machine users and non-users for demographic, socio-economic and institutional characteristics of households (Table 2). The average ages of household heads for renters and non-renters were 46.69 and 42.91 respectively. The result of the t-test showed that age of household heads had a significant mean difference between renters and non-renters at less than 1% probability level. The aged household heads were adopted agricultural machine renting better than the young aged. These maybe aged household heads have better experience and knowledge about the importance of agricultural machine renting than the young aged household heads. The overall household heads education level was an average of 3.78 years. The study result shows that the farmers who adopted agricultural machines were relatively more educated (with an average of 4.19 years and 3. 39 years for

**Table 2. Demographic, socio-economic, institutional and pilot characteristics of households by agricultural machine renting for continuous variable.**

| Description | Machine renting N = 192 | Not Machine renting N = 208 | Total | t-test |
|---|---|---|---|---|
| Age of household head | 46.69 (0.74) | 42.91 (0.61) | 44.72 (0.48) | 3.98*** |
| Education level | 4.19 (0.22) | 3.39 (0.21) | 3.78 (0.15) | 2.61*** |
| Family size | 4.86 (0.12) | 4.53 (0.11) | 4.69 (0.08) | −1.92** |
| Distance from market | 2.87 (0.13) | 3.48 (0.18) | 3.19 (0.12) | −2.55*** |
| Distance from road | 2.34 (0.11) | 2.4 (0.09) | 2.38 (0.09) | 0.35 |
| Distance from DA | 2.22 (0.09) | 2.35 (0.09) | 2.29 (0.06) | 1.01 |
| Extension visit | 12.4 (0.51) | 9.76 (0.35) | 11.03 (0.31) | 4.34*** |
| Total land | 2.41 (0.07) | 1.89 (0.05) | 2.14 (0.04) | 5.88*** |
| Own land holding | 1.6 (0.06) | 1.35 (0.05) | 1.47 (0.04) | 3.12*** |
| Number of oxen | 2.7 (0.07) | 2.1 (0.05) | 2.39 (0.04) | 6.43*** |
| Total livestock holding (TLU) excluding oxen | 4.18 (0.16) | 3.99 (0.15) | 4 (0.1) | −1.03 |

Note: ***, ** and * refers significant at 1, 5 and 10% significant levels, respectively. The value in parenthesis refers standard deviations (Std. Dev)

non- adopters) and the difference was statistically significant at less than 1% probability level. Family size was another variable that had a significant mean difference between renters and non-renters at less than 5% probability level.

Moreover, the result of the t-test revealed that total land and ownership of oxen had significant mean difference between renters and non-renters at less than 1% probability level. Household heads that adopted agricultural machine renting have a comparatively larger farm size and a greater number of oxen. The study result shows that farmers who adopted agricultural machine renting reported higher frequencies of contact with extension agent than non-adopters. Non-adopter farmers were located far from the nearest markets (3.48 Km) compared to adopters (2.87Km). Distance from the nearest market had a significant mean difference between agricultural machine renting renter and non-renters at less than 1% of significance level. This shows that farmers who lived nearest to market centers adopted agricultural machine renting than those who were located far from market centers.

The chi-square test results for categorical variables were given in Table 3. It indicated that power in social activities and use of credit had a significant mean difference between agricultural machine renting adopters and non-adopters at less than 1% probability level. About 63.59% of non-renters farmers had power in social activity whereas 36.41% of farmers were renters. Household heads who adopted agricultural machine renting had less power to do social activity compared to non-renters. In addition, the chi square test showed that farmers who adopted agricultural machine renting were fewer users of credit. Moreover, participation in off-farm activity and access to information from extension agent about agricultural machine renting had significant differences between agricultural machine renting renters and non-renters at less than 10% probability level.

### Agricultural machine renting

In the study area, tractors, combine harvestesrs and maize Sheller are used in the production process of cereal crops, either solely or simultaneously. The study showed that Table 4 approximatly 25.3% rented tractors only for ploughing services. Farmers commonly rent tractors in the study area for the first and second tillage. This is becauset the first ploughing operation is the heaviest and most challenging for oxen, whereas a tractor can plough easily and deeply. Similarly, about 22.3% of farmers rented combined harvesters only for wheat harvesting, threshing and winnowing operation and about 28.54% rented maize Sheller machines only for shelling of maize after harvesting. About 11.16% of farm households rent three types of machines (tractor, combine harvester and maize Sheller) simultaneously.

**Table 3. Demographic, socio-economic, institutional characteristics of households by agricultural machine renting for categorical variable.**

| Variables | Level | No. | % | No. | % | $x^2$-value |
|---|---|---|---|---|---|---|
| Sex | Male = 1 | 195 | 51.45 | 184 | 48.55 | 0.87 |
| Power on social | Yes | 71 | 36.41 | 124 | 63.59 | 37.0*** |
| Member-coop | Yes | 168 | 52.66 | 151 | 47.34 | 0.28 |
| Credit use | Yes | 72 | 41.62 | 101 | 58.38 | 13.16*** |
| Off-farm parti | Yes | 76 | 58.02 | 55 | 41.98 | 2.82* |
| Information EA | Yes | 155 | 54.96 | 127 | 45.04 | 3.37* |

Source: Computed from survey data (2021)

**Table 4. Major agricultural machine used in the study area.**

| Types of agricultural machines used | Freq. | Percent |
|---|---|---|
| Tractor | 118 | 25.3 |
| Combiner | 104 | 22.3 |
| Sheller | 133 | 28.54 |
| All (tractor, combiner and Sheller) | 52 | 11.16 |
| Tractor &combiner | 17 | 3.65 |
| Tractor & Sheller | 20 | 4.3 |
| Combiner & Sheller | 22 | 4.72 |

## Impacts of agricultural machine renting on productivity and commercialization

The essential tests under econometric analysis that verified the model were undertaken on hypothesized variables. The variables included in the model were tested for the problems of multicollinearity and heteroskedasaticity. The result shows that there is no problem of multicollinearity and heteroskedasaticity among the variables. Further, the model specification was carried out using the Ramsey-reset test and the results revealed that there were no omitted variables in the model.

Table 5 represents the first stage ESR binary probit estimation results. The probit model fits the data reasonably well [Wald Chi-squared = 137.31, p = 0.000)]. The model results reaveled that household, socioeconomic, and institutional factors had a significant impact on the decision to rent agricultural machines.

The age of the household head affected adoption of agricultural machine renting positively and significantly at 1% level. The marginal effect result indicates that aged household heads adopt more agricultural machines than younger households because, as the age of households increases, working power decreases. This finding is similar to [13]. Agricultural machine renting increased with the level of education of the household heads. Household heads who attened better education level have a good awareness about agricultural machines. Educated household heads are more likely to use agricultural machines [10,39]. The total land cultivated significantly and positively affected the adoption of agricultural machines. Farmers who owned large amount of cultivated land were more likely to adopt agricultural machines than small land-holding farmers. [63] found farm size was a positive influence on the adoption of agricultural technology packaging. Ownership of large amounts of cultivated land often required more labor and time to perform agricultural operations.

Frequency of extension contact influenced the adoption of agricultural machines significantly and positively at the 1% level. This implies that farmers who are frequently in contact with extension agents are more likely to adopt agricultural machines than are those in lower contact with extension agents. This is because, development agents provide advice on agricultural machine utilization and benefits to farmers. The extension provides in-depth information, training and consultancy services on the source, application and relevance of agricultural technologies for farmers. Similarly, farmers with

**Table 5. Decision of adopting agricultural machines renting: Probit model.**

| Variables | Coeff | Std.Err | Marginal Effect |
|---|---|---|---|
| Age of household | 0.03*** | 0.009 | 0.012 |
| Education level of household | 0.09*** | 0.026 | 0.036 |
| Adult equivalent | −0.11 | 0.059 | −0.045 |
| Poweron social activity | −0.49*** | 0.149 | −0.196 |
| Market distance | −0.07* | 0.043 | −0.031 |
| Road distance | 0.14** | 0.073 | 0.058 |
| Distance from office | 0.1271 | 0.086 | −0.050 |
| Extension visit | 0.0376*** | 0.012 | 0.015 |
| Member of cooperative | 0.2692 | 0.185 | 0.107 |
| Credit use | −0.4276*** | 0.146 | −0.170 |
| Total land | 0.3655*** | 0.040 | 0.145 |
| TLU | −0.0372 | 0.041 | −0.014 |
| Number of oxen | 0.3934*** | 0.110 | 0.156 |
| Off farm activities | −0.0666 | 0.158 | −0.026 |
| Information for mechanization | 0.3425** | 0.165 | 0.136 |
| -Cons- | −2.1487*** | 0.705 | |

Source: Computed from survey data (2021)

information about agricultural machines from government extension agents are more likely to adopt agricultural machines. This is because the provision of information helps farmers to become aware of the benefits and costs of machines. The use of credit negatively and significantly affected the probability of farmers using agricultural machines to produce cereal crops at 1%. The result indicated that farmers who used credit services decreased the probability of agricultural machines renting. This may be because households used credit for consumption, fattening livestock and home construction.

Distance from the market and the main road negatively and significantly affected the adoption of agricultural machines by 10% and 5%, a significant level. Proximity to the market and main roads are important factor in adopting agricultural machines because better access to markets and roads is an indicator of access to information and infrastructure. Thus, farmers who live near the market and main road are more likely to rent agricultural machines and other important inputs at a lower cost than those who live far away from markets and main roads. Similarly, [85] the adoption decision of smallholders affected by distance from the marketplace due to cost of production. The number of oxen affects the adoption of agricultural machines positively and significantly at 1%. This implies an increase in the number of oxen and the probability of adopting agricultural machines increases. This means that households with many oxen perform agricultural activities like the second plow, third plow, sowing and threshing more easily than thos fewer oxen.

Table 6 shows the results ATT and ATU for the key outcome variables related to agricultural machine renting. In this analysis the main outcome vaiables are TFP, TE and COIM index. Before, we analyzed the result of ESR model to check the validity of instrumental variables using a falsification test. The falsification test shows the instrumental variable significantly affects the adoption of agricultural machines but not TFP, TE and COIM index. Instrument variables are jointly statistically significant in the adoption equation [$\chi2 = 4.22$ (p = 0.04)] but not outcome functions. The instrumental variable did not affect the TFP [F = 0.10(p = 0.7497)] technical efficiency [F = 0.45 (p = 0.5038)], and commercialization index [F = 1.92 (p = 0.1668)].

The adoption of agricultural machines was renting increases total productivity, technical efficiency and commercialization level of farm households. For farmers who adopted agricultural machine renting but theoretically would have not adopted, then their TFP decreased by 0.573 (57.3%). This means, the ATT of total factor productivity of cereal crops

**Table 6. Endogenous switching regression model result (average treatment effects).**

| Outcome variables | Mean farm household type and treatment effect | Decision stage | | Treatment effect |
|---|---|---|---|---|
| | | To Adopt | Not to Non-adopt | |
| Productivity | ATT | 3.1 | 2.52 | 0.573*** |
| | ATU | 3.13 | 2.7 | 0.422*** |
| TE | ATT | 0.845 | 0.741 | 0.104*** |
| | ATU | 0.837 | 0.772 | 0.064*** |
| COIM | ATT | 0.572 | 0.452 | 0.120*** |
| | ATU | 0.544 | 0.444 | 0.099*** |

Note. ATT=Adoption effect for adopters, ATU=Adoption effect for non-adopters, and *** p<.01.

would decrease by 57.3% as a result of did not use agricultural machines renting. Likewise, if non-users of agricultural machine adopted agricultural machine renting, then their TFP would have significantly increased by 0.419(41.9%). For farmers who adopted agricultural machines but theoretically would not have adopted them, then their TE decreased by 0.104 (10.4). This means, the ATT of TE on cereal crops would decrease by 10.4% due to do not adopting agricultural machines. Similarly, if non-users of agricultural machine adopted agricultural machines, then their TE would have significantly increased by 0.064(6.4%). In addition, the results of the commercialization index of cereal crops for adopters would drop by 0. 12(12%) if the adopters were not adopted. In the same way, if the non-users of agricultural machine had adopted agricultural machine renting, their COIM would have increased by 0.098 (9.8%).

Consequently, ESR shows that in Ethiopia the adoption of agricultural machine renting would have increased productivity and commercialization of cereal crops. This result resembles to [3] the adoption of agricultural technology has a positive effect on crop productivity including wheat. The adoption of agricultural technologies has also a positive and significant impact on productivity of teff [84].

## Popensity score matching estimation results

In addition to the ESR model, this study uses the PSM technique to check the robustness of the results obtained from the ESR model. Propensity scores (the probability of adoption in agricultural machine) are estimated using a probit model. Fig 2 shows the distribution of adopter and non-adopter households with respect to estimated propensity scores. The figure illustrates the estimated propensity distribution for treatment and control households. The upper half of the graph refers to the propensity score distribution of treatment groups, while the bottom half shows the control groups. The y-axis refers to the densities of estimated propensity scores.

There should be a common support condition that should be applied to how the propensity score is distributed between adopters of agricultural machine renting and those who do not. The estimated propensity scores varied between 0.0826423 and 0.9633814 (mean=0. 6249002) and 0. 0347458 and 0.92732443 (mean,=0.3462459) in the treatment and control groups, respectively. The common support lies between 0.0826423 and 0.92732443. Table 7 presents the balancing tests for each matching algorithm before and after matching. The results show that the mean standardized bias is reduced after matching from 10.3 to 5.3 percent) compared to before matching (38percent). Similarly, Pseudo-$R^2$ declined substantially, from 22.4 percent to 1 percent. The p-values from the likelihood ratio tests showed that all of the covariates were statistically significant at a p-value of less than 1 percent prior to matching, but insignificant post-match. These p-values clearly demonstrate that the matching process balance the observed characteristics between treated and control after matching.

Table 8 shows the result of PSM, using three different matching algorithm techniques (nearest neighbor matching (NNM), Kernel based matching (KBM), and Radius matching methods). The result reveals that, as in the ESR analysis,

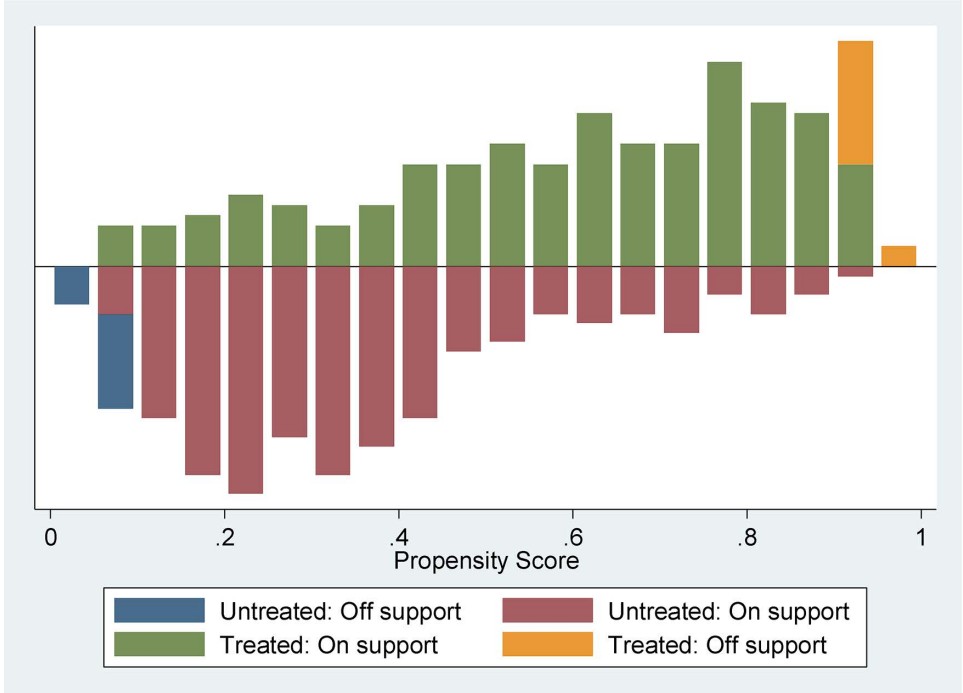

**Fig 2. Distribution of estimated propensity distribution for treatment and control groups and common support area.**

**Table 7. Covariates balancing tests before and after matching.**

| Matching algorism | Pesido-R-square | | LR$x^2$P−value | | Mean standard bias | |
|---|---|---|---|---|---|---|
| | **Before** | **After** | **Before** | **After** | **Before** | **After** |
| NNM$_1$ | 0.224 | 0.010 | 123.87 (0.00) | 4.99 (0.986) | 38 | 5.3 |
| NNM$_2$ | 0.224 | 0.014 | 123.87 (0.00) | 6.71 (0.278) | 38. | 6.9 |
| KBM$_3$ | 0.224 | 0.010 | 123.87 (0.00) | 4.83 (0.729) | 38 | 6.6 |
| KBM$_4$ | 0.224 | 0.011 | 123.87 (0.00) | 5.36 (0.980) | 38. | 6.5 |
| RCM$_5$ | 0.224 | 0.030 | 123.87 (0.00) | 8.55 (0.881) | 38 | 7.8 |
| RCM$_6$ | 0.224 | 0.030 | 123.87 (0.00) | 10.22 (0.746) | 38 | 10.3 |

NNM1 = One nearest neighbor matching and common support

NNM$_2$ = Two nearest neighbor matching and common support

KBM3 = Kernel with band width 0.05 and common support

KBM4 = Kernel with band width 0.1 and common support

RCM4 = Radius Caliper 0.05 matching

RCM5 = Radius Caliper 0.1 matching

**Table 8. Average treatment effects: propensity score matching.**

| Outcome variable | Matching algorism | Mean of outcome variables based on matched observation | | ATE | ATT | SE |
|---|---|---|---|---|---|---|
| | | Adopter | Non-adopter | | | |
| COMI | NNM₁ | 0.573 | 0.459 | 0.1105 | 0.114 | 0.015 |
| | NNM₂ | 0.573 | 0.459 | 0.105 | 0.113 | 0.016 |
| | KBM₃ | 0.573 | 0.46 | 0.105 | 0.113 | 0.016 |
| | KBM₄ | 0.573 | 0.459 | 0.105 | 0.113 | 0.016 |
| | RCM₅ | 0.555 | 0.456 | 0.098 | 0.100 | 0.013 |
| | RCM₆ | 0.567 | 0.451 | 0.108 | 0.116 | 0.011 |
| TFP | NNM₁ | 3.10 | 2.57 | 0.46 | 0.524 | 0.104 |
| | NNM₂ | 3.10 | 2.6 | 0.44 | 0.49 | 0.102 |
| | KBM₃ | 3.10 | 2.58 | 0.44 | 0.511 | 0.093 |
| | KBM₄ | 310 | 2.58 | 0.45 | 0.513 | 0.090 |
| | RCM₅ | 3.12 | 2.64 | 0.46 | 0.47 | 0.080 |
| | RCM₆ | 3.11 | 2.68 | 0.39 | 0.411 | 0.066 |
| TE | NNM₁ | 0.847 | 0.753 | 0.089 | 0.093 | 0.020 |
| | NNM₂ | 0.847 | 0.745 | 0.089 | 0.102 | 0.017 |
| | KBM₃ | 0.847 | 0.749 | 0.085 | 0.098 | 0.015 |
| | KBM₄ | 0.847 | 0.751 | 0.84 | 0.096 | 0.015 |
| | RCM₅ | 0.847 | 0.773 | 0.073 | 0.077 | 0.010 |
| | RCM₆ | 0.847 | 0.774 | 0.070 | 0.077 | 0.009 |

the adoption of agricultural machine renting resulted in an increase in TFP, TE and COIM. The PSM result revealed that increased adoption of agricultural machines renting significantly increases total factor productivity of crops in the range of 0.41 to 0.52 (41–52 percent). Similarly, the adoption of agricultural machines increases technical efficiency from 0.077 to 0.104 (7.7% −10.4%). Moreover, on average, the adoption of agricultural machines increased the household's commercialization level in the range from 0.102 to 0.116 (10 to 11.6 percent). Therefore, it can be concluded that, apart from the minor differences in magnitude between the PSM and ESR estimates, the results of ESR and PSM indicate that the adoption of agricultural machines had positive impacts on TFP, TE and COIM levels. This finding is similar to [79] the result from PSM and ESR estimates of soil and water conservation practice on food insecurity and net crop value.

## Summary

The agricultural sector in Ethiopia is the main source of economic development, and its growth originates from smallholder farmers. Even though smallolder agricultural production and productivity still low. Thus, this study analyzed the of impact agricultural mahines on the productivity and commercialization of cereal crops in West Gojjam, Ethiopia. A series of formal and informal surveys were carried out during the 2020/2021 crop season on 400 sample farm household heads that were randomly drawn from six kebeles in three woredas through a multi-stage sampling procedure. Secondary data were collected from published and unpublished sources to support the primary data.

Descriptive statistics and econometric models were used to analyze the data. The descriptive statistics include the mean, percentage, t-test and chi-squre tests. Econometric models, including endogenous switch regression (ESR) and propensity score matching (PSM) were employed.

The descriptive statistics showed that approximately 48% sample of households were renting agricultural machines to perform different agricultural operations. The study also investigates the total factor productivity and technical efficiency of cereal crops by using the total factor productivity index and stochastic frontier function. The results showed that the total

factor index of the cereal crop was 2.89 and the stochastic frontier model showed that technical efficiency of cereal crop producers was 80.6%. The impact of agricultural machine renting on productivity and commercialization was estimated using endogenous switch regression and propensity score matching models. In these models, 192 machine users and 208 non-users of agricultural machines were used. The results of the first ESR model showed that age, education level, ownership of oxen, total cultivated land, extension visit, access to information, family size, the position a farmer, and distance to the nearest market significantly affected adoption of agricultural machine renting. The results of the endogenous switch regression and propensity score matching model indicate that the adoption of agricultural machines has positive and significant impacts on total factor productivity, technical efficiency and output commercialization index.

## Policy implication

The adoption of agricultural machines has a positive and significant impact on the productivity and commercialization of cereal crops. This result indicates that agricultural mechanization enhances productivity and commercialization of cereal crops in the study area. When agricultural mechanization increases productivity and commercialization, we should examine what factors determine agricultural machine renting. Based on the result of study determined by demographic, socio-economic and institustional factors. Therefore, governmental and non-governmental stakeholders give more emphasis on:

- formulating appropriate policies to provide adequate and effective basic educational opportunities for rural farming households.
- deliver selective and appropriate agricultural machines to smallholder farmers
- creating better extension services and offering practical training or demonstration on the utilization of agricultural machines for plough, harvesting, threshing and shelling of crops.
- strengthening the existing rural-urban market centers, rural-urban roads and other infrastructure development activities in the study area.

Finally, the government also should formulate a supporting policy for local importers of agricultural implements and strengthen their capacity.

## Supporting information

**S1 Data.**
(DOCX)

**S1 File.**
(XLSX)

## Author contributions

**Conceptualization:** Selam Tilahun, Berhanu Kuma, Amsalu Bedemo.

**Data curation:** Berhanu Kuma, Amsalu Bedemo.

**Formal analysis:** Selam Tilahun.

**Investigation:** Selam Tilahun.

**Methodology:** Selam Tilahun.

**Supervision:** Berhanu Kuma, Amsalu Bedemo.

**Validation:** Berhanu Kuma, Amsalu Bedemo.

**Visualization:** Berhanu Kuma, Amsalu Bedemo.

**Writing – original draft:** Selam Tilahun.

**Writing – review & editing:** Selam Tilahun.

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
