## [Decision Letter · Decision Letter 0]

7 Aug 2024

Dear Dr. Tilahun,

Thank you for submitting your manuscript to PLOS ONE. After careful consideration, we feel that it has merit but does not fully meet PLOS ONE’s publication criteria as it currently stands. Therefore, we invite you to submit a revised version of the manuscript that addresses the points raised during the review process.

We look forward to receiving your revised manuscript.

Kind regards,

Cataldo Pulvento

Academic Editor

PLOS ONE

Reviewers' comments:

Reviewer's Responses to Questions

**Comments to the Author**

1. Is the manuscript technically sound, and do the data support the conclusions?

Reviewer #1: Yes

Reviewer #2: Yes

2. Has the statistical analysis been performed appropriately and rigorously?

Reviewer #1: Yes

Reviewer #2: Yes

3. Have the authors made all data underlying the findings in their manuscript fully available?

Reviewer #1: Yes

Reviewer #2: Yes

4. Is the manuscript presented in an intelligible fashion and written in standard English?

Reviewer #1: Yes

Reviewer #2: Yes

Reviewer #1: Impacts of Agricultural Machine Renting on Cereal Crop Productivity and Commercialization in West Gojjam Zone, Ethiopia

Reviewer comments:

The paper's overall structure needs work. There is no research gap in the abstract, and it is rather superficial. The introduction briefly explains the necessity of undertaking this research but does not provide a clear picture of the investigation.

Before the methodology, it should be necessary to conduct a literature review, both theoretical and empirical, and the authors should clarify how the conceptual framework guides the study.

The area of the description part is shallow, and it needs more explanation in terms of agricultural machines renting on cereal crop productivity and commercialization in West Gojjam Zone, particularly in your selected districts.

To determine the sample size, you should not show the sampling techniques and procedures. Which random sampling techniques were employed to select the sample respondents? What are the benchmarks—192 households adopted and 208 households non-adopted? Please support it with empirical evidence. From Table 1, how did you calculate the sample households?

From the data collection methods, you stated that the major data collection methods included observations and visual aids. I have not seen photos in the discussion part. What is important for setting the data collection method?

The methods of data analysis, particularly the descriptive statistics part, were not clearly explained, and why not consider other alternative methods like inferential statistics (chi-square test and t-test)? The chi-square test and t-test should be used to determine statistical differences and the association between agricultural machine adopters and non-adopters on independent variables.

Under the econometric model, you should separately explain the theoretical and analytical framework, and then you should specify the model and the conditional expectations, treatment, and heterogeneity effect.

Measurement of market participation: It is so shallow; it needs more explanation with supporting theoretical and empirical literature reviews.

Under equation 4 the authors stated input factors (labour, land, fertilizer seed and herbicide) for production function. In addition to these variables, why not incorporated agricultural machines (like tractor, combiner, and Sheller) as inputs for the production of cereal crops?

The impacts of agricultural machine rental on productivity and commercialization: The specification of the endogenous switching regression model should not be well organized. So, please specify the ESR model while supporting the theoretical and empirical literature review.

Under the definition of variables, not only list the independent variables; you should define each independent variable and hypothesize based on the literature review.

Under the discussion part, specifically the descriptive results, the authors simply show the number rather than the implication of the numbers. What are the implications of your results or numbers (percent, mean, standard deviation, min, and max)?

Under the discussion part, specifically the impacts of agricultural machine renting on productivity and commercialization, the investigation needs deeper and more detailed discussion of the implications of the findings and how to relate them to the existing literature.

The paper's major contributions and their consequences for practice and policy are not succinctly summarized in the conclusion, which comes off as hurried. It is necessary to have a more thorough conversation about proposed policies.

Reviewer #2: 1. What is the contribution of this study to the existing literature?

2. Adding the description, measurement and expected hypothesis of the independent variable in the methodology section with citation.

3. If the farmers are used more than two machines, why not use multinomial logit model?

4. What are the diagnostic tests that you conducted?

5. Support your finding by more empirical evidence.

6. Policy implication is too general, so make it specific to the concerned body.

7. Generally this paper lacks of consistency.

**Do you want your identity to be public for this peer review?** For information about this choice, including consent withdrawal, please see our Privacy Policy

Reviewer #1: No

Reviewer #2: No

---

## [Author Response · Author response to Decision Letter 1]

4 Sep 2024

We appreciate the constructive and insightful comments of both reviewers for their careful reading of the manuscript and their constructive remarks, on our manuscript entitled ‘‘Impacts of Agricultural Machine Renting on Cereal Crop Productivity and Commercialization in West Gojjam Zone, Ethiopia". We are pleased by the comment, input, and clarity raised by both reviewers’. We have taken the reviewers’ comments fully to incorporate, clarify, and improve our manuscript, which will meet the standard of the Plos ONE

---

## [Decision Letter · Decision Letter 1]

11 Jun 2025

Dear Dr. Tilahun,

Thank you for submitting your manuscript to PLOS ONE. After careful consideration, we feel that it has merit but does not fully meet PLOS ONE’s publication criteria as it currently stands. Therefore, we invite you to submit a revised version of the manuscript that addresses the points raised during the review process.

We look forward to receiving your revised manuscript.

Kind regards,

Miquel Vall-llosera Camps

Senior Staff Editor

PLOS ONE

**Journal Requirements:**

Reviewers' comments:

Reviewer's Responses to Questions

**Comments to the Author**

Reviewer #1: All comments have been addressed

2. Is the manuscript technically sound, and do the data support the conclusions?

Reviewer #1: Yes

3. Has the statistical analysis been performed appropriately and rigorously?

Reviewer #1: Yes

4. Have the authors made all data underlying the findings in their manuscript fully available?

Reviewer #1: Yes

5. Is the manuscript presented in an intelligible fashion and written in standard English?

Reviewer #1: Yes

**Reviewer #1: ** As you can see from the reviewer's main document, the writers address the majority of the comments; nonetheless, they did not address some of them.

**Do you want your identity to be public for this peer review?** For information about this choice, including consent withdrawal, please see our Privacy Policy

Reviewer #1: No

---

## [Author Response · Author response to Decision Letter 2]

9 Aug 2025

Thank you for constructive comments and suggestions.

---

## [Decision Letter · Decision Letter 2]

1 Sep 2025

Impacts of Agricultural Machine renting on Cereal Crop Productivity and Commercialization in West Gojjam Zone, Ethiopia.

PONE-D-24-08983R2

Dear Dr. Selam Tilahun,

We’re pleased to inform you that your manuscript has been judged scientifically suitable for publication and will be formally accepted for publication once it meets, on the one hand the minor revisions suggested by reviewers 1 and 3 and, on the other hand, all outstanding technical requirements.

Kind regards,

Serge Svizzero, Ph.D

Academic Editor

PLOS ONE

Additional Editor Comments (optional):

Reviewer #1:

Still two questions were not addressed: "What are the benchmarks for the 192 households that adopted and the 208 households that did not adopt? Please support it by empirical evidence.""In addition to labor, land, fertilizer, seed, and herbicide variables, why not incorporate agricultural machines (like tractors, combines, and shellers) as inputs for the production of cereal crops?"

Reviewer #3:

As I have seen both the original and revised paper (PONE-D-24-08983R2), the paper is original and creative and covers machine renting, a relatively recent Ethiopian farming practice. The paper presented valuably research output by considering critical gaps in the literature and can be used in the future as a reference source in the newly developing field of study in Ethiopia.

1. The Study Presents the Results of Original Research

The paper reports a new result using data from three potential districts in west Gojjam Zone for the stated technology renting practice. The finding is supported with adequate and reliable data according to the statement of the research ethics, and the study focussed on Ethiopian farming section which is hot issue i.e., agricultural machine renting.

2. Results reported have not been published elsewhere

The results are primarily new and logical. The method of sample determination and sampling producers are scientific and concrete. Yet, there is a bit idea repetition in Table 5: Major agricultural machine renting in the study area (from the previous authors published paper: determinant of adoption of agricultural machine renting in West Gojjam zone, Ethiopia). So, minor sections of the methods and Table 4.Major agricultural machine used in the study area should be modified by authors to improve the paper quality. If Table 5 adopted from the previous paper the author should add citation under Table 4. This will make the paper contribution to be more scientific, concrete and readable.

3. Experiments, statistics, and other analyses are performed to a high technical standard and are described in sufficient detail.

Statistical analyses and validity tests are done well and presented in a clear order. The paper would improve more by adding hypothesis testing prior to each models result interpretation. Additionally, a table summarizing each variable, definition of the variable, unit of the variable, expected sign of the variable would make the paper more clear and scientific rather than putting in sentence form.

4. Conclusions are presented in an appropriate fashion and are supported by the data.

The conclusions are relevant and justified by the data. The implications of policy are well-framed and timely in the Ethiopian mechanized farming. It can benefit more by focusing on main results and detailing the scientific basis of relationships amid variables in a logical order. Moreover, inclusion of study limitations and future research direction will enhance the paper further in terms of quality and replication.

5. The article is presented in an intelligible fashion and is written in Standard English.

The writing is well-structured and clear overall but needs again some readability improvements and typing errors and technical terms improving.

6. The research meets all applicable standards for the ethics of experimentation and research integrity.

The study follows to ethical standards and research integrity. So, minor revisions to the structure of headings and subheadings numbering is required to ensure comprehensive flow of idea more.

7. The article adheres to appropriate reporting guidelines and community standards for data availability.

The presentation or attachment of the raw cross-section data in excel is notable. It gives support to the strength of transparency, replication, and validate the results.

Recommendation: the paper is suitable for publication by considering the proposed revisions specifically paraphrasing idea redundancy, clarifying hypotheses, sharpening conclusions, and readability issues as stated above.

Reviewers' comments:

Reviewer's Responses to Questions

**Comments to the Author**

Reviewer #1: All comments have been addressed

Reviewer #3: (No Response)

2. Is the manuscript technically sound, and do the data support the conclusions?

Reviewer #1: Yes

Reviewer #3: Yes

3. Has the statistical analysis been performed appropriately and rigorously?

Reviewer #1: Yes

Reviewer #3: Yes

4. Have the authors made all data underlying the findings in their manuscript fully available?

Reviewer #1: Yes

Reviewer #3: Yes

5. Is the manuscript presented in an intelligible fashion and written in standard English?

Reviewer #1: Yes

Reviewer #3: Yes

Reviewer #1: Still two questions were not addressed: "What are the benchmarks for the 192 households that adopted and the 208 households that did not adopt? Please support it by empirical evidence.""In addition to labor, land, fertilizer, seed, and herbicide variables, why not incorporate agricultural machines (like tractors, combines, and shellers) as inputs for the production of cereal crops?"

Reviewer #3: Review report (date: 31.8.2025)

Impacts of Agricultural Machine renting on Cereal Crop Productivity and Commercialization in West Gojjam Zone, Ethiopia with Manuscript Number: PONE-D-24-08983R2

General issues (considering seven guiding principles): As I have seen both the original and revised paper (PONE-D-24-08983R2), the paper is original and creative and covers machine renting, a relatively recent Ethiopian farming practice. The paper presented valuably research output by considering critical gaps in the literature and can be used in the future as a reference source in the newly developing field of study in Ethiopia.

1. The Study Presents the Results of Original Research

The paper reports a new result using data from three potential districts in west Gojjam Zone for the stated technology renting practice. The finding is supported with adequate and reliable data according to the statement of the research ethics, and the study focussed on Ethiopian farming section which is hot issue i.e., agricultural machine renting.

2. Results reported have not been published elsewhere

The results are primarily new and logical. The method of sample determination and sampling producers are scientific and concrete. Yet, there is a bit idea repetition in Table 5: Major agricultural machine renting in the study area (from the previous authors published paper: determinant of adoption of agricultural machine renting in West Gojjam zone, Ethiopia). So, minor sections of the methods and Table 4.Major agricultural machine used in the study area should be modified by authors to improve the paper quality. If Table 5 adopted from the previous paper the author should add citation under Table 4. This will make the paper contribution to be more scientific, concrete and readable.

3. Experiments, statistics, and other analyses are performed to a high technical standard and are described in sufficient detail.

Statistical analyses and validity tests are done well and presented in a clear order. The paper would improve more by adding hypothesis testing prior to each models result interpretation. Additionally, a table summarizing each variable, definition of the variable, unit of the variable, expected sign of the variable would make the paper more clear and scientific rather than putting in sentence form.

4. Conclusions are presented in an appropriate fashion and are supported by the data.

The conclusions are relevant and justified by the data. The implications of policy are well-framed and timely in the Ethiopian mechanized farming. It can benefit more by focusing on main results and detailing the scientific basis of relationships amid variables in a logical order. Moreover, inclusion of study limitations and future research direction will enhance the paper further in terms of quality and replication.

5. The article is presented in an intelligible fashion and is written in Standard English.

The writing is well-structured and clear overall but needs again some readability improvements and typing errors and technical terms improving.

6. The research meets all applicable standards for the ethics of experimentation and research integrity.

The study follows to ethical standards and research integrity. So, minor revisions to the structure of headings and subheadings numbering is required to ensure comprehensive flow of idea more.

7. The article adheres to appropriate reporting guidelines and community standards for data availability.

The presentation or attachment of the raw cross-section data in excel is notable. It gives support to the strength of transparency, replication, and validate the results.

Recommendation: the paper is suitable for publication by considering the proposed revisions specifically paraphrasing idea redundancy, clarifying hypotheses, sharpening conclusions, and readability issues as stated above.

---

## [Editor Report · Acceptance letter]

PONE-D-24-08983R2

PLOS ONE

Dear Dr. Tilahun,

I'm pleased to inform you that your manuscript has been deemed suitable for publication in PLOS ONE. Congratulations! Your manuscript is now being handed over to our production team.

Kind regards,

on behalf of

Pr. Serge Svizzero

Academic Editor

PLOS ONE